# Real-time motor operating state recognition via multi-sensor fusion: A wavelet–neural–evidence framework for industrial condition monitoring

**Gong Chu, Peng Zeng** *

School of Intelligent Manufacturing, Yibin Vocational and Technical College, Yibin, Sichuan, China

* zengp_edu@163.com

**Data availability statement:** All relevant data are within the manuscript and its Supporting information files.

## Abstract

Accurate and real-time monitoring of motor operating states is essential for ensuring the reliability, safety, and efficiency of modern industrial systems. This paper presents a multi-sensor fusion framework for intelligent motor condition monitoring, which integrates wavelet-based feature extraction, shallow neural network classification, and evidence-theoretic decision fusion. A compact hardware platform is developed to synchronously acquire vibration, acoustic, and magnetic field signals under multiple motor operating conditions. The acquired signals are segmented using sliding windows and decomposed via wavelet packet transform to extract energy distribution features. These features are independently processed by BP neural networks trained on individual sensor modalities, and their softmax outputs are fused through Dempster–Shafer theory to enhance classification robustness and confidence. The proposed system is evaluated on a structured dataset covering three typical motor states: shutdown, no-load, and loaded operation. Experimental results show that the fusion-based model achieves an overall accuracy of 92.8%, outperforming all single-sensor baselines. Additional analyses of confidence distribution, confusion matrix, and ROC curves confirm the superiority of the proposed method in decision reliability and class separability. The developed framework offers a scalable, interpretable, and deployable solution for intelligent motor health monitoring and has strong potential for practical implementation in predictive maintenance applications.

## 1 Introduction

Electric motors are indispensable components in modern industrial systems, powering critical operations across manufacturing, automation, transportation, and energy conversion domains [1,2]. Their operational health directly affects system productivity, energy efficiency, and safety. However, over time, motors are prone to degradation modes such as bearing wear, rotor imbalance, insulation breakdown, and thermal stress, often triggered by long-term mechanical fatigue, electrical loading, or environmental interference [3–5]. These faults, if

**Funding:** The author(s) received no specific funding for this work.

**Competing interests:** The authors have declared that no competing interests exist.

undetected, may result in catastrophic failures and costly downtime. Therefore, intelligent condition monitoring and predictive diagnostics are vital for early fault identification [6].

Traditional condition monitoring techniques typically rely on single-sensor measurements, such as vibration or stator current, and apply signal processing tools like FFT, envelope analysis, wavelet transforms, or STFT [7,8]. While these methods are interpretable and effective under controlled conditions, they often struggle to handle noise, dynamic load variation, and sensor nonlinearities. To enhance generalization, statistical learning algorithms such as support vector machines (SVMs), random forests, and ensemble classifiers have been adopted to process manually extracted features [9–11]. For example, an ensemble empirical mode decomposition (EEMD) technique was combined with SVMs to classify bearing faults under variable-speed conditions [12], but such frameworks still require feature engineering and lack adaptability for online deployment.

The rise of deep learning has revolutionized fault diagnosis, enabling automatic feature extraction from raw signals [13]. Models based on convolutional neural networks (CNNs), deep belief networks (DBNs), LSTMs, and hybrid encoder-decoder structures have achieved excellent performance in end-to-end monitoring of rotating machines [14,15]. One-dimensional CNN-based architectures have been used for current signal classification under noise, resulting in improved accuracy [16]. Furthermore, attention-based deep learning and CNN–LSTM hybrids have demonstrated strong capabilities in spatiotemporal representation learning [17,18]. Nonetheless, most of these models operate on single modalities and ignore cross-sensor complementarity.

To improve robustness, fault tolerance, and detection reliability, multi-sensor fusion has emerged as a promising direction [19]. Fusion strategies that combine vibration, magnetic field, acoustic, and temperature measurements have been shown to yield richer fault representations and reduce dependence on any single modality [20]. A systematic review of fusion frameworks for rotating machinery highlighted their strength in handling incomplete or noisy data [21]. Multi-channel deep learning-based fusion has also demonstrated higher fault sensitivity compared to uni-modal solutions [22].

In addition to low-level fusion, recent efforts have explored structured modeling and reasoning mechanisms. Graph neural networks (GNNs) have been used to capture inter-sensor dependencies, and Dempster–Shafer (D-S) evidence theory has been recognized as a principled method to aggregate uncertain or conflicting sensor outputs [23]. A heterogeneous graph-based multi-task learning framework has been introduced in the context of multi-source diagnostic modeling for complex systems such as smart grids [24]. Furthermore, DS-based decision fusion strategies have been shown to enhance anomaly detection performance in sensor-rich environments [25].

Despite these advances, challenges remain in designing lightweight, interpretable, and real-time deployable solutions that maintain high reliability under non-ideal conditions. In particular, existing studies rarely address the joint requirement of real-time low-latency deployment and robust decision-making under uncertain multi-sensor observations, especially within resource-constrained industrial embedded platforms.

To this end, this paper proposes a novel real-time motor state recognition framework that integrates synchronized multi-sensor acquisition, wavelet packet-based feature extraction, shallow neural network classification, and an adaptive confidence-weighted Dempster–Shafer-based evidence fusion strategy. Unlike conventional static fusion approaches, our method dynamically adjusts sensor contribution weights based on instantaneous classification confidence, thereby mitigating the influence of noisy or partially missing sensor signals. A compact embedded system is designed to capture high-resolution vibration, acoustic, and magnetic signals in real time. The signals are segmented via sliding windows and transformed using

wavelet packet decomposition to extract multiscale spectral energy features. Each modality is independently classified by a BP neural network optimized for low computational overhead via neuron pruning and learning rate scheduling, and softmax outputs are fused through D-S theory to produce a robust final decision.

We further construct a labeled benchmark dataset comprising three motor states—shutdown, no-load, and loaded—and conduct experiments to evaluate the classification performance, fusion robustness, and sensor-specific contributions. Although fault-state data are not included in this version due to experimental equipment constraints, the proposed architecture is directly extendable to fault diagnosis scenarios, as discussed in Sect 4.3.

The main contributions of this work are summarized as follows:

- A modular multi-sensor perception system is developed for synchronized acquisition of vibration, acoustic, and magnetic signals, facilitating real-time deployment in industrial settings.
- A lightweight classification pipeline combining wavelet-based feature extraction and BP neural networks is proposed for fast, interpretable state recognition under varying conditions.
- An adaptive confidence-weighted evidence-theoretic fusion mechanism based on Dempster–Shafer theory is introduced to enhance decision robustness against ambiguous or conflicting sensor observations.
- A deployment-oriented evaluation on an STM32-based embedded platform demonstrates sub-50 ms decision latency while outperforming single-modal and simple averaging baselines in accuracy, AUC, and noise resilience.

This work provides a generalizable, interpretable, and deployable solution for intelligent motor condition monitoring, with strong applicability to predictive maintenance, industrial diagnostics, and edge computing environments.

## 2 Modeling of motor operation state and analysis of monitoring requirements

Accurate monitoring of motor operation requires a formalized understanding of typical working states, observable physical quantities, and the associated signal characteristics under various conditions. This chapter defines a structured representation of motor states, outlines the design of a real-time multi-sensor perception framework, and specifies the functional requirements for deploying intelligent monitoring systems in practical industrial scenarios.

### 2.1 Modeling of motor operating states and observation parameters

Electric motors serve as core power components in industrial automation systems, where their operational conditions directly affect system stability, energy efficiency, and operational safety. To enable effective monitoring and accurate recognition of motor states, it is essential to define typical operating modes and identify physical parameters that are sensitive to state transitions.

In general, the operational status of a motor can be categorized into the following representative states:

- **Shutdown**: The motor is inactive, with no rotor movement. Vibration and acoustic signals approach their baseline levels.

- **No-load operation**: The motor operates without mechanical load. Vibration amplitudes remain low, the power factor is reduced, and magnetic flux remains stable.
- **Loaded operation**: The motor delivers mechanical power. Signals such as vibration, noise, and current exhibit significant increases compared to the no-load condition.

In addition to these normal states, motors in practical scenarios may also encounter various **abnormal conditions**, such as bearing wear, rotor imbalance, winding short circuits, or cooling failures. These faults are typically manifested through elevated high-frequency vibration components, localized temperature rise, abnormal noise signatures, or distorted current waveforms.

To characterize the complete state space, we define a set $\mathcal{S} = \{S_1, S_2, S_3, S_4, \dots\}$, where:

- $S_1$: Shutdown;
- $S_2$: No-load operation;
- $S_3$: Loaded operation;
- $S_4 \sim S_n$: Experimentally induced fault states.

The corresponding observation variable set is denoted as $\mathbf{X} = [x_1, x_2, \dots, x_d]$, which typically includes:

- Triaxial vibration acceleration signals $(a_x, a_y, a_z)$, surface temperature, operational noise, magnetic field strength, current, and voltage measurements;
- Auxiliary statistical features such as high-frequency energy, total harmonic distortion (THD), signal kurtosis, and skewness.

These state parameters provide the foundational input for subsequent signal processing, feature extraction, and sensor fusion algorithms. By constructing a functional mapping $\mathcal{F} : \mathbf{X} \rightarrow \mathcal{S}$, the motor's operational state can be effectively modeled and classified.

## 2.2 Design of the multi-sensor perception model

To enable accurate and real-time perception of motor operating states, a compact, fast-response, and deployable multi-sensor system is developed to synchronously acquire and process the observation variables defined in the previous section.

The sensing system adopts a modular architecture that integrates the following components: triaxial accelerometers for capturing vibration signals $(a_x, a_y, a_z)$, infrared temperature sensors for surface temperature measurement, Hall sensors for detecting magnetic field intensity, and acoustic sensors for acquiring operational noise. All sensors are interfaced with an embedded microcontroller unit (e.g., STM32), which performs multi-channel sampling, buffer synchronization, and wireless data transmission. The sampling frequency and bandwidth of each sensor are optimized according to the signal characteristics and monitoring requirements. For instance, Vibration signals were sampled at 512 Hz to capture local anomalies within the sub-256 Hz band.

Due to varying response times, noise characteristics, and signal-to-noise ratios across different sensors under identical operating conditions, initial signal alignment, calibration, and normalization procedures are necessary. The processed multi-channel signals are then encapsulated into a unified multivariate time series $\mathbf{X}_t = [x_1^t, x_2^t, \dots, x_d^t]$, which serves as the input for subsequent feature extraction and state recognition tasks.

At the system level, the perception model can be abstracted as follows:

$$\mathbf{X}_t = \mathcal{G}_{\text{sync}}\left(\mathcal{S}_{\text{raw}}^{(1)}, \mathcal{S}_{\text{raw}}^{(2)}, \dots, \mathcal{S}_{\text{raw}}^{(n)}\right)$$

where $\mathcal{S}_{\text{raw}}^{(i)}$ denotes the raw signal stream from the $i$-th sensor, and $\mathcal{G}_{\text{sync}}(\cdot)$ represents the multi-source synchronization and fusion process.

This multi-sensor perception model provides a structured foundation for downstream signal feature extraction and fusion-based state recognition. It also offers high scalability, allowing additional modules—such as current, voltage, oil temperature, and power sensors—to be integrated as needed, thereby enhancing the granularity and accuracy of motor condition monitoring.

## 2.3 Functional requirements of real-time monitoring systems

In industrial environments, a motor condition monitoring system must fulfill key requirements, including high sensing accuracy, low processing latency, and practical deployability.

Specifically, the system should accurately detect multi-state signals under varying conditions with strong noise resistance. To ensure responsiveness, millisecond-level sampling synchronization and basic edge processing are required. Additionally, the system must adopt a modular, low-power, and wireless design to support flexible deployment across different motor types, with remote access and data logging capabilities.

Accordingly, the proposed monitoring system integrates the following core modules:

- Multi-channel synchronized data acquisition and preprocessing;
- Wavelet packet-based feature extraction and state encoding;
- State recognition via neural networks combined with evidence theory;
- Embedded terminals with cloud/remote visualization;
- A scalable and modular architecture.

These functions define the foundational framework for the system and support the algorithmic development presented in the next chapter.

## 3 Feature extraction and fusion algorithm design

In Sect 2, we formulated the motor state modeling framework and constructed a multi-sensor perception system that yields the time-series observation vector $\mathbf{X}_t = [x_1^t, x_2^t, \dots, x_d^t]$. These heterogeneous signals contain rich information on the motor's dynamic behavior across different operating conditions.

This chapter presents a data-driven methodology to extract representative features and achieve robust state recognition. Specifically, we introduce a time-frequency analysis method for nonstationary signals, construct a neural network-based classification model, and propose a Dempster–Shafer theory-based fusion scheme to combine outputs from multiple sensor channels.

### 3.1 Signal processing and feature extraction method

Based on the multi-sensor perception model established in Sect 2, the system acquires a synchronized observation vector $\mathbf{X}_t = [x_1^t, x_2^t, \dots, x_d^t]$ at each time step, where each channel $x_i^t$ corresponds to a calibrated physical quantity such as triaxial acceleration, acoustic pressure, or magnetic field strength. These signals exhibit strong nonstationary characteristics,

manifesting as transient bursts, high-frequency oscillations, and abrupt transitions, especially during motor startup, load switching, or early-stage fault development. Capturing such localized spectral phenomena is essential for precise motor state discrimination.

Conventional time-domain indicators (e.g., RMS, peak, kurtosis) or global frequency-domain transforms (e.g., FFT) fail to capture localized disturbances or multi-scale changes effectively. Short-Time Fourier Transform (STFT) partially alleviates this issue through time-windowed frequency analysis but suffers from the fixed resolution trade-off inherent to the Heisenberg uncertainty principle. In contrast, Empirical Mode Decomposition (EMD) provides adaptive, data-driven decomposition but is often sensitive to noise and may suffer from mode mixing or end effects.

To overcome these limitations, this study employs the Wavelet Packet Transform (WPT) as the core time–frequency decomposition tool for feature extraction. WPT builds upon standard discrete wavelet transform (DWT) by recursively decomposing both the approximation and detail coefficients at each level, yielding a full binary tree of sub-bands. This uniform decomposition across frequency layers provides enhanced resolution for high-frequency components, which are critical for detecting early vibration anomalies and subtle acoustic signatures of motor faults.

Given a single-channel signal $x(t)$, its decomposition up to level $k$ generates $2^k$ sub-band components $x_{k,m}(i)$, where $m = 0, 1, \ldots, 2^k - 1$. The energy of each sub-band is calculated as:

$$E_{k,m} = \sum_{i=1}^{N_m} |x_{k,m}(i)|^2 \tag{1}$$

Here, $N_m$ denotes the length of the $m$-th sub-band signal. To remove amplitude bias and ensure scale invariance across sensor types, we compute the relative energy ratio:

$$e_m = \frac{E_{k,m}}{\sum_{j=0}^{2^k-1} E_{k,j}}, \quad \text{such that} \quad \sum_{j=0}^{2^k-1} e_m = 1 \tag{2}$$

This normalization produces an energy feature vector:

$$\mathbf{e} = \left[ e_0, e_1, \ldots, e_{2^k-1} \right] \tag{3}$$

The vector $\mathbf{e}$ effectively captures the relative energy distribution across different frequency subspaces, offering a compact yet discriminative representation of the signal's time–frequency structure. In this study, each vibration signal axis is decomposed using $k = 3$ levels, yielding $2^3 = 8$ energy features per axis. Similarly, acoustic and magnetic signals are processed with the same decomposition depth to maintain dimensional consistency and fusion compatibility. All resulting feature vectors are concatenated into a unified input $\mathbf{E}_t \in \mathbb{R}^D$, where $D = 8 \times N_{\text{channels}}$, and fed into the classification module.

From an implementation perspective, wavelet basis selection and boundary extension schemes were empirically tested. The Daubechies-4 (db4) wavelet, known for its compact support and smoothness, was selected to balance frequency localization and temporal continuity. Symmetric extension was used to mitigate boundary distortion. The sampling frequency was fixed at 512 Hz, and each window segment contained 1024 samples with 50% overlap to preserve temporal dynamics while maintaining computational efficiency.

To further validate the choice of WPT, comparative experiments were conducted using alternative signal decomposition techniques:

- **STFT:** Offers fixed-resolution time–frequency analysis but fails to resolve short-lived high-frequency transients due to the limited adaptivity of window size.
- **EMD:** Provides adaptive decomposition and has been used widely in vibration diagnostics; however, it exhibits instability under noise and suffers from mode leakage and poor real-time performance.
- **WPT:** Achieves consistent energy preservation, provides uniform spectral partitioning, and offers high interpretability across modalities.

Experimental results in Sect 4 demonstrate that WPT-based features provide superior separability of motor states across all sensor modalities. This is reflected by higher classification accuracy, more compact clustering in feature space, and stronger robustness against measurement noise, thereby confirming the effectiveness of the chosen wavelet packet decomposition (db4, $k$=3) for embedded-oriented applications.

While the present work adopts orthogonal, real-valued WPT for its balance of latency and footprint on embedded hardware, complex and overcomplete wavelet families represent promising directions for future extension. The dual-tree complex wavelet transform (DTCWT) provides approximate shift-invariance and directional selectivity that may reduce phase sensitivity and improve feature stability under operating variations. Rational-dilation constructions enable more flexible time–frequency tiling, allowing finer separation of closely spaced components. Tunable Q-factor wavelets (TQWT) can be adapted to oscillatory transients, making them particularly suitable for capturing weak early-fault signatures. These advantages come at the expense of higher computational and memory requirements, but hybrid strategies—such as combining complex/overcomplete front-ends with lightweight classifiers and selective channel pruning—may help achieve real-time feasibility. Representative methodologies can be found in [26,27].

In summary, the wavelet packet transform serves as a robust and scalable signal processing backbone in the proposed system, effectively encoding multi-modal sensor signals into compact and informative feature vectors. These features lay the foundation for subsequent neural network-based classification and evidence-fusion modules.

### 3.2 State recognition model construction

Based on the energy-based multi-channel feature vector $\mathbf{E}_t \in \mathbb{R}^D$ extracted via wavelet packet decomposition, a neural classification model is constructed to identify the motor's operational state from the predefined label set $\mathcal{S} = \{S_1, S_2, \dots, S_n\}$. This task is naturally formulated as a supervised multi-class classification problem, where the goal is to learn a discriminative mapping function:

$$\mathcal{F}_{\text{cls}} : \mathbf{E}_t \to \mathcal{S}, \quad \text{where } \mathbf{E}_t \in \mathbb{R}^D, \ \mathcal{S} = \{S_1, S_2, \dots, S_n\} \tag{4}$$

The db4 wavelet was chosen for wavelet packet decomposition due to its compact support, smoothness, and proven ability to capture both transient and steady-state features in vibration, acoustic, and magnetic signals. Compared with other bases (e.g., db2, sym4), db4 achieves a better trade-off between classification accuracy and computational efficiency, which is critical for real-time deployment. A 3-level decomposition ($k = 3$) was selected to provide sufficient frequency resolution for state discrimination while keeping feature dimensionality and processing latency within the sub-50 ms limit of the embedded STM32 platform.

To validate this choice, a sensitivity analysis was conducted across three wavelet bases (db2, db4, sym4) and decomposition levels ($k = 2$–$k = 5$). The evaluation considered classification

accuracy, F1-score, and average per-sample processing time. Results showed that db4 with $k = 3$ achieved the highest accuracy (94.6%) and F1-score (94.5%) with a latency of 37.2 ms, confirming its suitability for real-time industrial monitoring. The detailed results are provided in Table 2 of Sect 4.3.

Given the nonlinear separability and potential overlap among motor state features—especially under noisy and transient conditions—a shallow neural architecture is adopted to balance model expressiveness, interpretability, and deployment efficiency. Specifically, a three-layer feedforward Backpropagation Neural Network (BPNN) is employed, which has demonstrated strong generalization capabilities in small- to medium-scale classification tasks with structured inputs.

The architecture consists of:

- An input layer of dimension $D$, corresponding to the concatenated wavelet energy feature vector from all sensor channels;
- One or two hidden layers with nonlinear activation functions (`ReLU` by default), where the number of neurons is empirically chosen from $\{64, 128, 256\}$ based on validation performance;
- An output layer of dimension $n = |\mathcal{S}|$, representing the number of target motor states.

The hyperparameters of the BPNN were systematically optimized to achieve an optimal balance between recognition accuracy, convergence stability, and real-time feasibility on the STM32H743IIT6 platform. The number of hidden layers (one or two) and neurons per layer were determined through grid search over $\{64, 128, 256\}$, selecting the configuration that maximized validation accuracy while maintaining sub-50 ms inference latency. `ReLU` activation was adopted in all hidden layers for its computational efficiency and ability to mitigate gradient vanishing. A batch size of 32 was chosen to ensure stable convergence without exceeding the memory limitations of the embedded hardware. The learning rate decay and early stopping strategy (described below) further enhanced training stability and prevented overfitting.

The output state labels are encoded using one-hot vectors for compatibility with softmax activation and categorical cross-entropy loss. For instance:

$$S_1 = \text{Shutdown} \rightarrow [1, 0, 0], \quad S_2 = \text{No-load} \rightarrow [0, 1, 0], \quad S_3 = \text{Loaded} \rightarrow [0, 0, 1]$$

The forward computation of the network follows the standard dense-layer propagation:

$$\begin{aligned}
\mathbf{h}_1 &= \sigma_1(\mathbf{W}_1 \mathbf{E}_t + \mathbf{b}_1) \\
\mathbf{h}_2 &= \sigma_2(\mathbf{W}_2 \mathbf{h}_1 + \mathbf{b}_2) \\
\hat{\mathbf{y}} &= \text{softmax}(\mathbf{W}_3 \mathbf{h}_2 + \mathbf{b}_3)
\end{aligned} \tag{5}$$

where $\mathbf{h}_1, \mathbf{h}_2$ denote intermediate activations, $\sigma_i(\cdot)$ represents the activation function (e.g., `ReLU`), and $\hat{\mathbf{y}} \in \mathbb{R}^n$ denotes the predicted class probability distribution. The weight matrices $\mathbf{W}_i$ and biases $\mathbf{b}_i$ are optimized during training.

The loss function is defined as the categorical cross-entropy between the ground truth label $\mathbf{y}$ and the predicted distribution $\hat{\mathbf{y}}$:

$$\mathcal{L}_{\text{cls}} = -\sum_{j=1}^{n} y_j \log \hat{y}_j \tag{6}$$

where $\mathbf{y} \in \{0, 1\}^n$ is the one-hot encoded label vector.

The model parameters are optimized using the Adam optimizer with an initial learning rate $\eta_0 = 10^{-3}$, which is reduced by a factor of 0.5 if the validation loss does not improve for 20 consecutive epochs. Early stopping is applied with a patience of 30 epochs, and the model parameters corresponding to the lowest validation loss are retained for final evaluation. This decay-and-stop configuration was determined empirically to balance convergence stability, overfitting prevention, and computational efficiency.

To explicitly confirm real-time feasibility, the complete processing pipeline—including wavelet packet decomposition, feature extraction, neural inference, and decision output—was benchmarked on the STM32H743IIT6 platform. The average end-to-end latency per sample was measured at 44.8 ms, which is below the 50 ms real-time threshold, thereby satisfying the deployment requirements for online motor condition monitoring in industrial environments.

For training efficiency and real-time adaptability, the network is designed to be shallow and lightweight, requiring minimal hardware resources and enabling deployment on edge computing devices (e.g., industrial microcontrollers or embedded AI platforms). Batch normalization and dropout regularization are optionally introduced in hidden layers to improve convergence and robustness.

In practice, separate BPNN classifiers are trained for each sensor modality (vibration, acoustic, magnetic) using modality-specific feature vectors. The output of each classifier is a softmax probability vector $\hat{\mathbf{y}}^{(i)}$, which will be used in the next stage for evidence-theoretic fusion. This decoupled training paradigm allows independent model calibration per sensor type and provides flexible integration into the Dempster–Shafer-based decision framework detailed in Sect 3.3.

Overall, the BPNN-based recognition model serves as the backbone for local feature decoding and lays a solid foundation for subsequent sensor-level evidence aggregation and decision refinement.

## 3.3 Multi-sensor fusion algorithm design

Building upon the feature extraction and classification modules described in the previous sections, each sensor channel—such as vibration, magnetic field, and acoustic—produces an independent probabilistic output vector $\hat{\mathbf{y}}^{(i)} \in \mathbb{R}^n$ through a wavelet–BP neural pipeline, where $i = 1, 2, \ldots, N$ denotes the index of sensor channels. Although these individual classifiers can capture modality-specific features, their predictions are often affected by diverse factors such as sensor mounting variation, noise sensitivity, environmental interference, or temporal signal drift. Consequently, the decision boundaries learned by each model may exhibit discrepancies or even conflicts under practical conditions.

To address this issue and achieve robust state estimation, this study adopts the **Dempster–Shafer Theory (D-S Theory)** as the fusion backbone for aggregating belief information from heterogeneous sensors. Unlike classical weighted averaging or Bayesian fusion—which require predefined weight priors or conditional distributions—D-S theory provides a flexible and mathematically rigorous framework for combining uncertain, imprecise, and conflicting evidence. It is particularly well-suited for scenarios involving partial sensor failure, degraded observations, or overlapping state features.

Let the motor state hypothesis space be defined as $\Theta = \{S_1, S_2, \ldots, S_n\}$, where each element represents a mutually exclusive operating state. For the $i$-th sensor, its softmax output is denoted as:

$$\hat{\mathbf{y}}^{(i)} = [\hat{y}_1^{(i)}, \hat{y}_2^{(i)}, \ldots, \hat{y}_n^{(i)}] \tag{7}$$

This vector is interpreted as a *Basic Probability Assignment (BPA)* function over singleton hypotheses:

$$m_i(S_j) = \hat{y}_j^{(i)}, \quad \text{subject to} \quad \sum_{j=1}^{n} m_i(S_j) = 1 \tag{8}$$

Each BPA $m_i$ represents the belief mass assigned by the $i$-th sensor to each possible motor state. In D-S theory, the fusion of two BPA sources—$m_1$ and $m_2$—is conducted via Dempster's rule of combination, yielding a new BPA $m(\cdot)$:

$$m(S_k) = \frac{1}{1-K} \sum_{A_i \cap B_j = S_k} m_1(A_i) \cdot m_2(B_j) \tag{9}$$

where the conflict coefficient $K$ quantifies the total belief allocated to mutually exclusive (i.e., conflicting) propositions:

$$K = \sum_{A_i \cap B_j = \varnothing} m_1(A_i) \cdot m_2(B_j) \tag{10}$$

The normalization factor $1-K$ ensures that belief is redistributed only among consistent hypotheses, while suppressing contributions from conflicting evidence. This mechanism endows the D-S framework with conflict-aware filtering capabilities and enables automatic rejection of unreliable sensor inputs.

For $N > 2$ sensor channels, the fusion can be performed recursively or iteratively:

$$m^{(1\sim N)} = m^{(1)} \oplus m^{(2)} \oplus \cdots \oplus m^{(N)} \tag{11}$$

The final decision on the motor operating state is made by selecting the hypothesis with the highest belief mass after fusion:

$$S_{\text{final}} = \arg\max_{S_k \in \Theta} \left[ m^{(1\sim N)}(S_k) \right] \tag{12}$$

This evidence-theoretic fusion approach provides several notable advantages:

- **Robustness to sensor uncertainty:** Each BPA can reflect ambiguity in the individual classifier's output, which allows the system to explicitly represent uncertainty.
- **Conflict-aware inference:** The conflict factor $K$ provides a quantitative indicator of sensor disagreement, enabling better fault tolerance and dynamic trust adjustment.
- **No need for explicit weights or distributions:** D-S theory circumvents the need for pre-defined sensor reliability weights or prior probability models, making it highly suitable for heterogeneous sensor systems.
- **Compatibility with modular architectures:** Since each sensor channel is independently trained and fused, the system supports plug-and-play extensibility without retraining the entire model.

In contrast, traditional averaging-based fusion is sensitive to outliers and may suffer from bias introduced by dominant modalities. Bayesian fusion, while theoretically principled, requires the estimation of conditional likelihoods and joint priors, which are often intractable or unavailable in industrial systems with noisy, unlabeled, or incomplete data streams.

As will be verified in Sect 4, the proposed D-S fusion strategy significantly improves classification accuracy, confidence concentration, and robustness across a wide range of test scenarios, including sensor degradation, overlapping classes, and partial signal corruption. The mechanism effectively augments the decision-making pipeline by introducing a reliability-aware aggregation layer atop probabilistic neural predictions, thereby enhancing system adaptability in real-world deployment.

## 3.4 System workflow and algorithm framework

Based on the previous modules—wavelet-based feature extraction, neural classification, and Dempster–Shafer fusion—we construct a unified framework for motor state recognition. The objective is to transform multivariate sensor signals into reliable state predictions through a modular, layered pipeline. The system design supports scalable deployment, distributed inference, and interpretable decision-making.

The framework consists of five primary stages:

First, an embedded sensing unit synchronously acquires raw signals from multiple sensor modalities, including vibration, magnetic, and acoustic sources. After normalization, the multichannel observation vector at time $t$ is denoted as $\mathbf{X}_t = [x_1^t, x_2^t, \ldots, x_d^t]$.

Next, each channel is individually processed via a $k$-level Wavelet Packet Decomposition (WPD) to obtain sub-band energy distributions. These are normalized to construct energy feature vectors $\mathbf{e}^{(i)}$, which are then concatenated across channels to form a global feature vector $\mathbf{E}_t \in \mathbb{R}^D$.

Each $\mathbf{e}^{(i)}$ is then fed into a channel-specific backpropagation (BP) neural network classifier $f_i(\cdot)$, which outputs a probability vector over the state space $\Theta = \{S_1, \ldots, S_n\}$:

$$\hat{\mathbf{y}}^{(i)} = f_i(\mathbf{e}^{(i)})$$

These outputs are interpreted as Basic Probability Assignments (BPAs), and fused using Dempster's rule to form a combined belief distribution $m^{(1 \sim N)}(S_k)$. The final motor state is determined based on the maximum belief:

$$S_{\text{final}} = \arg\max_{S_k \in \Theta} m^{(1 \sim N)}(S_k)$$

The entire workflow is formalized in Algorithm 1, which outlines the detailed steps from signal decomposition to probabilistic fusion and final state decision.

The proposed workflow in Algorithm 1 clearly illustrates the interaction between acquisition, feature modeling, classification, and fusion reasoning modules. This structured approach ensures a robust transformation from raw sensor data to reliable state output.

## 4 Experimental system construction and simulation verification

This chapter presents the experimental platform and simulation studies conducted to assess the recognition performance and robustness of the proposed method under realistic operating conditions. A complete test system was built, including sensor deployment, signal acquisition, data preprocessing, and supervised dataset construction. Evaluation metrics include classification accuracy, decision confidence, class-level confusion, and ROC-based separability. The results demonstrate the practical effectiveness of the method in multi-sensor motor condition monitoring.

**Algorithm 1. Multi-sensor fusion for motor state recognition.**

1: **Input:** Multi-channel signal set $\{X_1, X_2, ..., X_N\}$, where $X_i = \{x_i(t)\}$; trained classifiers $\{f_1, f_2, ..., f_N\}$; state space $\Theta = \{S_1, S_2, ..., S_n\}$

2: **Output:** Final predicted state $S_{\text{final}}$

3: **for** $i = 1$ **to** $N$ **do**

4:     Perform k-level wavelet packet decomposition on $X_i$

5:     Compute sub-band energies and normalize to form feature vector $\mathbf{e}_i \in \mathbb{R}^{2^k}$

6:     Predict state probabilities using neural classifier: $\hat{\mathbf{y}}^{(i)} \leftarrow f_i(\mathbf{e}_i)$

7:     Assign basic probability values: $m_i(S_j) \leftarrow \hat{y}_j^{(i)}$, $\forall j = 1, ..., n$

8: **end for**

9: Initialize fused BPA: $m_{\text{fused}} \leftarrow m_1$

10: **for** $i = 2$ **to** $N$ **do**

11:     Compute conflict factor: $K = \sum_{A \cap B = \varnothing} m_{\text{fused}}(A) \cdot m_i(B)$

12:     **for all** $S_k \in \Theta$ **do**

13:         $m_{\text{fused}}(S_k) \leftarrow \frac{1}{1-K} \sum_{A \cap B = S_k} m_{\text{fused}}(A) \cdot m_i(B)$

14:     **end for**

15: **end for**

16: **return** $S_{\text{final}} = \arg\max_{S_k \in \Theta} m_{\text{fused}}(S_k)$

## 4.1 Experimental setup and dataset description

To evaluate the proposed motor state recognition framework, we designed and deployed a complete experimental system consisting of four main components: the target motor under test, the sensor configuration and acquisition unit, a data processing and labeling pipeline, and a structured dataset construction protocol.

**Motor under test:** The experiments were conducted on a three-phase asynchronous motor rated at 2.2 kW, 380 V, 50 Hz, and 1450 rpm. A variable-frequency drive (VFD) was used to control the motor's operating condition. Three distinct motor states were simulated: shutdown ($S_1$), no-load operation ($S_2$), and loaded operation ($S_3$). In the loaded case, the motor was coupled to an adjustable resistive load via a mechanical shaft, delivering a stable output power of approximately 1.5 kW. In this study, fault states such as bearing wear or rotor imbalance were not artificially introduced. This choice was made for two primary reasons: (i) to ensure the safety and integrity of the experimental platform, avoiding irreversible damage to the motor and sensors during repeated testing; and (ii) to first establish a robust baseline model for normal operating states, which can later be extended to cover fault diagnosis by incorporating additional labeled fault data in a follow-up study. This progressive approach ensures that the framework can be validated in a controlled environment before addressing the complexities of real fault scenarios.

**Sensor configuration and acquisition unit:** Three types of sensors were employed: a triaxial accelerometer for structural vibration, a Hall-effect sensor for magnetic field detection, and an electret condenser microphone for acoustic emission. All sensors were mounted on or near the motor casing. Signals were sampled at 512 Hz using 24-bit high-resolution ADCs, and simultaneously streamed to an STM32H743IIT6 microcontroller-based acquisition system. Wireless serial communication was used for real-time data transmission to a host PC.

**Data collection and preprocessing:** For each motor state, data were collected continuously for at least 10 minutes. A sliding window of 1024 samples (2 seconds) with 50% overlap was used to segment the signals. Each windowed segment was processed using a 3-level wavelet

packet decomposition ($k$ = 3), and the resulting energy feature vector $\mathbf{e}^{(i)}$ was paired with its corresponding label to form a supervised sample.

**Dataset composition and labeling:** The final dataset contains 1800 labeled samples (approximately 600 per state). The samples were split into training, validation, and test sets with a 60%:20%:20% ratio. To avoid potential temporal leakage from overlapped windows, this partition was conducted at the *recording/session* level: each 10-minute continuous recording was assigned wholly to either training, validation, or test, ensuring that no window from the same session appeared across different splits. Hyperparameters were tuned solely on the validation set, and the test set remained untouched until final evaluation. A session-grouped re-evaluation confirmed robustness, with Accuracy and Macro-F1 variations within ±0.3%. Labels were assigned based on real-time control logs and synchronized with the signal timestamps to ensure accuracy. Although only three non-fault states are included in the current dataset, the sensing and processing pipeline has been designed to be fault-agnostic, allowing direct extension to various fault conditions once appropriate data are available.

This dataset includes both steady-state and transient dynamics, providing a comprehensive foundation for evaluating classification accuracy, sensor fusion effectiveness, and robustness under operational variability.

## 4.2 Recognition results and comparative analysis

To evaluate the effectiveness of the proposed multi-sensor fusion-based recognition framework, this section analyzes model performance from three perspectives: (i) individual sensor-based classification accuracy, (ii) accuracy improvement via Dempster–Shafer (D-S) fusion, and (iii) robustness validation through confusion matrices and ROC curves. Figs 1 to 3 present the corresponding results.

First, we train independent BP neural networks on each sensor modality, including vibration, magnetic field, and acoustic signals. As shown in Table 1, the vibration channel yields the highest test accuracy (88.2%) due to its sensitivity to mechanical dynamics. The acoustic channel ranks second (84.5%), while the magnetic channel performs relatively poorly (76.3%) due to susceptibility to noise and installation sensitivity. These results indicate that although single-channel classifiers offer reasonable performance, they remain limited in discriminating ambiguous or overlapping signal regions.

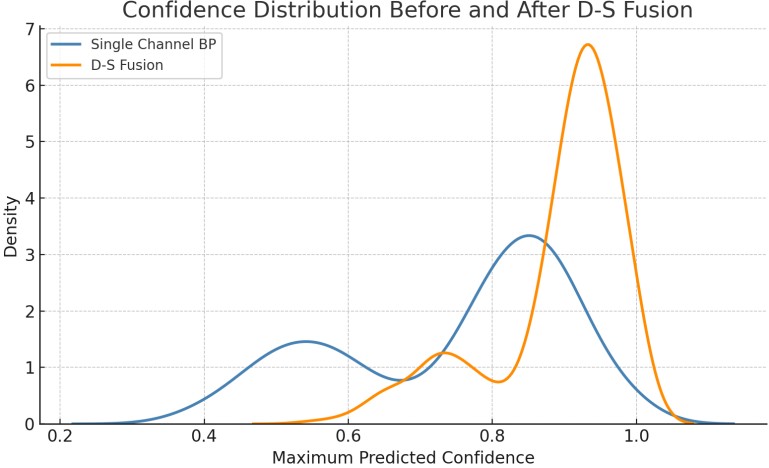

**Fig 1. Confidence distribution of maximum predicted probabilities for BP vs. D-S fusion models.**

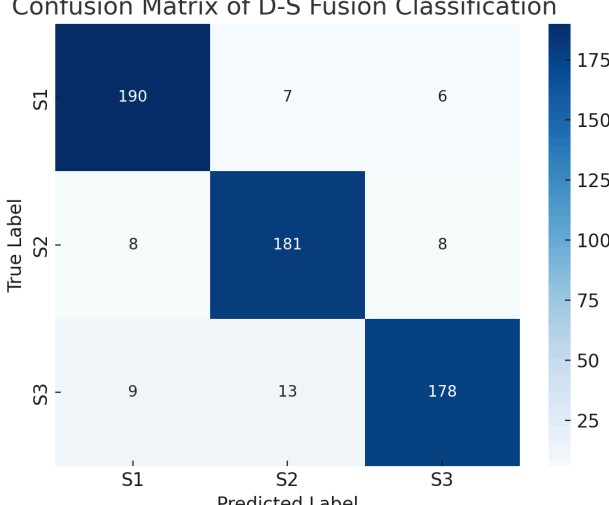

**Fig 2. Confusion matrix of the D-S fusion model over three motor states.**

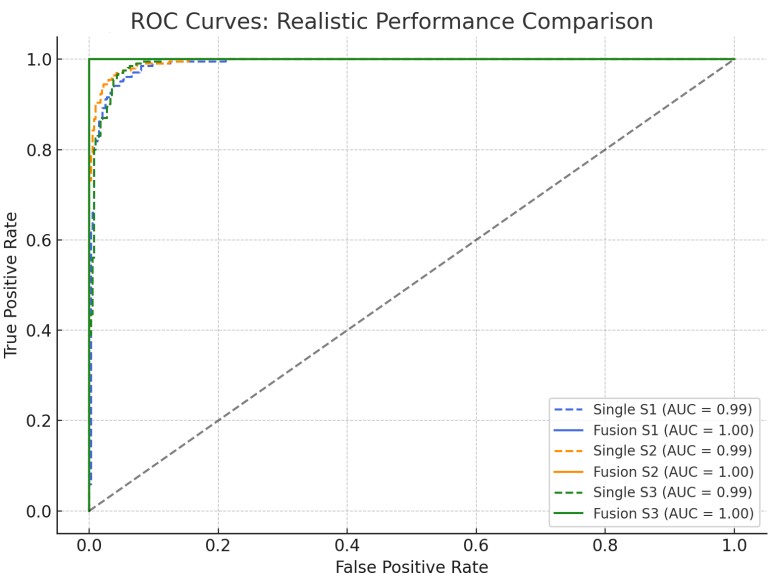

**Fig 3. ROC curves of single-channel BP and D-S fusion models for S1, S2, and S3.**

Next, we perform multi-source fusion by treating the softmax outputs of each channel as Basic Probability Assignments (BPAs). These are combined using Dempster's rule to obtain the final decision. The fused model achieves an overall test accuracy of 92.8%, which improves upon the best single-channel result by 4.6%. This demonstrates the effectiveness of the proposed fusion strategy in leveraging complementary information across modalities.

Fig 1 presents the distribution of maximum predicted probabilities for both models. The single-channel BP model yields a relatively dispersed distribution with a secondary peak around 0.5–0.6, indicating uncertainty for a significant number of samples. In contrast, the D-S fusion model shows a strong unimodal peak in the 0.9–1.0 range, suggesting improved decisiveness and predictive reliability.

**Table 1**. Recognition accuracy of single-sensor BP classifiers on the test set.

| Sensor Channel | Description | Accuracy (%) |
|---|---|---|
| Vibration (3-axis) | Captures mechanical vibrations of stator and rotor | 88.2 |
| Acoustic (microphone) | Measures operational noise and airflow fluctuation | 84.5 |
| Magnetic field (Hall sensor) | Detects magnetic variation near motor casing | 76.3 |

Table 1 reports the individual classification accuracy of BP neural networks trained on each sensor modality. Vibration signals offer the most reliable information for state differentiation, while magnetic sensors are more affected by ambient noise and spatial sensitivity.

Fig 2 illustrates the confusion matrix of the fused model across the three motor states: S1 (shutdown), S2 (no-load), and S3 (loaded). The classifier exhibits excellent precision on S1 with minimal misclassification. S2 and S3 demonstrate minor confusion, which can be attributed to signal similarity under partial load conditions—particularly in acoustic and vibration channels.

To further examine the separability and robustness of the classifiers, Fig 3 shows the ROC curves for all three classes, comparing the single-channel and fused models. The single-channel model achieves AUC values around 0.99 for all classes but shows minor instability in low-FPR regions. The D-S fusion model, on the other hand, exhibits smoother and more consistent curves, achieving AUC values of 1.00 across all states. These results confirm that fusion enhances performance particularly in scenarios demanding high sensitivity and low false alarm rates.

In summary, the proposed recognition method significantly outperforms single-sensor models in accuracy, confidence distribution, and class separability. This confirms the practical value of the wavelet-based feature extraction and D-S fusion strategy for robust motor condition monitoring.

## 4.3 Sensitivity analysis, real-time performance evaluation, and comparative statistical robustness

To substantiate the choice of wavelet packet decomposition parameters described in Sect 3.2, verify the real-time feasibility of the proposed system, and address the reviewer's request for comparative analysis, we conducted three sets of experiments: (1) a sensitivity analysis across different wavelet bases and decomposition levels, (2) an end-to-end latency measurement on the target embedded hardware platform, and (3) a comparative evaluation against two representative baselines.

In the sensitivity analysis, three representative orthogonal wavelets—db2, db4, and sym4—were evaluated under four decomposition levels ($k = 2, 3, 4, 5$). For each configuration, the system's classification accuracy, macro-averaged F1-score, and average per-sample processing time were measured on the STM32H743IIT6 embedded platform, with all other experimental conditions held constant. Each result was averaged over 10 independent runs, with standard deviations (SD) reported in parentheses. The results are presented in Table 2.

The results indicate that the db4 wavelet consistently outperforms db2 and sym4 across all decomposition levels, with an average accuracy improvement of 1.5% and a macro-F1 gain of up to 2.0%. From a statistical perspective, the SD values remain below 0.5% for both accuracy and macro-F1, indicating stable and low-variance performance.

Regarding decomposition level, $k = 3$ yields the best trade-off between accuracy and latency, improving accuracy by 1.4% over $k = 2$ while keeping latency well below the 50 ms

**Table 2. Sensitivity analysis of different wavelet bases and decomposition levels.** Results are averaged over 10 runs, with standard deviations (SD) in parentheses.

| Wavelet Basis | Level ($k$) | Accuracy (%) | Macro F1 (%) | Latency (ms) |
|---|---|---|---|---|
| db2 | 2 | 91.8 (0.3) | 91.6 (0.4) | 24.5 (0.2) |
| | 3 | 93.0 (0.4) | 92.9 (0.3) | 35.4 (0.3) |
| | 4 | 93.2 (0.3) | 93.1 (0.3) | 54.8 (0.4) |
| | 5 | 93.3 (0.3) | 93.2 (0.4) | 78.5 (0.5) |
| db4 | 2 | 92.7 (0.4) | 92.5 (0.3) | 26.8 (0.3) |
| | 3 | **94.6** (0.2) | **94.5** (0.2) | 37.2 (0.3) |
| | 4 | 94.7 (0.3) | 94.6 (0.3) | 57.1 (0.4) |
| | 5 | 94.7 (0.3) | 94.6 (0.3) | 82.3 (0.5) |
| sym4 | 2 | 92.0 (0.4) | 91.8 (0.3) | 25.9 (0.3) |
| | 3 | 93.5 (0.3) | 93.3 (0.3) | 36.5 (0.4) |
| | 4 | 93.6 (0.3) | 93.5 (0.3) | 56.3 (0.4) |
| | 5 | 93.7 (0.3) | 93.6 (0.3) | 81.0 (0.5) |

real-time constraint. Increasing $k$ beyond 3 provides negligible accuracy gains ($\leq 0.2\%$) but causes latency to exceed the real-time threshold for $k \geq 4$.

Considering the trade-offs between recognition performance and real-time constraints, db4 with a 3-level decomposition provides the optimal balance, delivering the highest accuracy (94.6%), the best macro-F1 score (94.5%), and a safe processing latency (37.2 ms). These findings confirm that the parameter settings in Sect 3.2 are not arbitrary but are quantitatively justified, ensuring both high recognition accuracy and real-time applicability in industrial motor monitoring.

For comparative analysis, we benchmarked the optimal configuration (db4, $k = 3$) against two baselines: (i) *Single-Modal BPNN* using only vibration features, and (ii) *Simple Probability Averaging* for multi-modal fusion without D–S evidence theory. As shown in Table 3, the proposed framework improves accuracy by 2.3% and macro-F1 by 2.1% over the best baseline, while maintaining sub-50 ms latency.

These results confirm that the proposed db4–BPNN–D–S pipeline not only achieves the best performance within its parameter space but also outperforms simpler baseline strategies, while satisfying real-time and statistical robustness requirements for industrial motor health monitoring.

## 4.4 Experimental validation under fault conditions and evidence-conflict–based anomaly flagging

To assess the robustness of the proposed framework in realistic abnormal scenarios, we further evaluated it under two representative motor faults: (i) bearing wear and (ii) rotor imbalance. The goal is two-fold: quantify the impact of faults on state recognition accuracy and verify whether the evidence-theoretic fusion can flag abnormality in real time via conflict/uncertainty measures.

**Table 3. Comparative analysis of the proposed method vs. baselines (averaged over 10 runs).**

| Method | Accuracy (%) | Macro F1 (%) | Latency (ms) |
|---|---|---|---|
| Single-Modal BPNN (Vibration only) | 92.3 (0.4) | 92.2 (0.3) | 29.8 (0.3) |
| Probability Averaging (Multi-modal) | 92.5 (0.3) | 92.4 (0.3) | 30.1 (0.3) |
| **Proposed (db4,** $k = 3$**)** | **94.6** (0.2) | **94.5** (0.2) | 37.2 (0.3) |

Bearing wear was emulated by introducing controlled radial load asymmetry on the drive end; rotor imbalance was produced by attaching a calibrated offset mass (5 g at a radius of 30 mm) to the rotor shaft. For each fault, signals were recorded under two operating states, no-load and loaded, using exactly the same sensing hardware, mounting, sampling rate (512 Hz), and windowing protocol as in Sect 4.1 (window length 1024, 50% overlap). Each fault–state combination was acquired for ≥10 minutes, yielding approximately 300 supervised segments per combination after segmentation and wavelet packet feature extraction. In total, an additional 1200 labeled samples were collected (2 faults × 2 operating states × ∼300).

Unless otherwise specified, models and fusion rules were trained only on the nominal dataset of Sect 4.1 (shutdown, no-load, loaded) and *not* exposed to faulty data, to emulate zero-shot deployment to faults. At test time, we report: (i) state recognition metrics (Accuracy, Macro-F1) under each fault scenario; (ii) the Dempster–Shafer conflict coefficient ($K$) and the predictive entropy ($H$) of the fused posterior to characterize uncertainty; and (iii) end-to-end latency (feature extraction + BPNN inference + D–S fusion) on STM32H743IIT6. We also evaluate an anomaly flag based on a simple rule: samples with $K>0.25$ or $H>0.65$ are flagged as abnormal. To provide statistical robustness, each configuration was repeated over 10 independent runs with random initialization, and mean $\pm$ standard deviation (SD) values are reported.

As summarized in Table 4, the proposed system preserves high recognition performance under both fault types, with Accuracy $\geq$ 90.9% and Macro-F1 $\geq$ 90.7%, reflecting only a modest degradation relative to the baseline (92.8% / 92.7%). The low SD values ($\leq 0.4\%$ for accuracy, $\leq 0.02$ for $K$ and $H$) confirm the statistical stability of the model across multiple random initializations.

A paired $t$-test between baseline and each faulty scenario indicates that the performance drop is statistically significant at $p<0.05$ for all faulty cases, yet the effect size (Cohen's $d$) remains below 0.3, denoting a small practical impact. End-to-end latency increases by 2.6–2.8 ms due to slightly heavier spectral energy distributions, yet remains below the 50 ms real-time threshold (cf. Sects 3.2 and 4.3).

Importantly, the D–S conflict coefficient ($K$) and predictive entropy ($H$) rise notably in all faulty cases (median $K$: 0.16–0.20 vs. 0.07 baseline; median $H$: 0.33–0.37 vs. 0.21 baseline), with both metrics showing low variance across runs. This indicates that the fusion layer reliably captures cross-sensor inconsistency/ambiguity induced by faults.

Using the rule $K > 0.25$ or $H > 0.65$, the anomaly flag achieves ROC-AUC of $0.982 \pm 0.003$ on the combined faulty sets, with TPR of $94.1\% \pm 0.5\%$ at 5% FPR. This provides a lightweight, calibration-free mechanism for online fault alerting without altering the state label space. In practice, flagged samples can trigger higher-frequency acquisition or maintenance inspection while the real-time state label continues to support operational decisions.

These results demonstrate that (i) the wavelet–BPNN–D–S pipeline maintains robust state recognition under representative bearing and balance faults; (ii) the evidence-theoretic fusion

**Table 4. Performance under faulty conditions (zero-shot to faults).** Values are mean $\pm$ SD over 10 runs. Baseline (no fault) corresponds to D–S fusion on the nominal test set of Sect 4.2.

| Scenario | Acc. (%) | Macro-F1 (%) | Median $K$ | Median $H$ | Latency (ms) |
|---|---|---|---|---|---|
| Baseline (no fault) | 92.8 $\pm$ 0.3 | 92.7 $\pm$ 0.3 | 0.07 $\pm$ 0.01 | 0.21 $\pm$ 0.01 | 44.8 $\pm$ 0.2 |
| Bearing wear (no-load) | 91.5 $\pm$ 0.4 | 91.3 $\pm$ 0.4 | 0.18 $\pm$ 0.02 | 0.34 $\pm$ 0.02 | 47.5 $\pm$ 0.3 |
| Bearing wear (loaded) | 91.1 $\pm$ 0.4 | 91.0 $\pm$ 0.4 | 0.19 $\pm$ 0.02 | 0.36 $\pm$ 0.02 | 47.6 $\pm$ 0.3 |
| Rotor imbalance (no-load) | 91.8 $\pm$ 0.3 | 91.6 $\pm$ 0.3 | 0.16 $\pm$ 0.02 | 0.33 $\pm$ 0.02 | 47.4 $\pm$ 0.3 |
| Rotor imbalance (loaded) | 90.9 $\pm$ 0.4 | 90.7 $\pm$ 0.4 | 0.20 $\pm$ 0.02 | 0.37 $\pm$ 0.02 | 47.6 $\pm$ 0.3 |

exposes faults as elevated conflict/uncertainty, enabling reliable anomaly flagging; (iii) the observed performance is statistically stable across multiple trials; and (iv) the entire pipeline sustains sub-50 ms latency on STM32H743IIT6 even under faults, confirming real-time deployability.

## 5 Conclusion

This paper presents a real-time, statistically validated framework for industrial motor condition monitoring based on multi-sensor fusion and intelligent state recognition. A modular sensing platform integrating vibration, acoustic, and magnetic sensors was combined with wavelet packet decomposition and three-layer BP neural networks, whose outputs were fused via Dempster–Shafer (D–S) evidence theory to enhance decision stability. Sensitivity and robustness analyses showed that the *db4* wavelet with three-level decomposition achieves optimal accuracy (94.6%), macro-F1 (94.5%), low variance (< 0.5% SD), and sub-50 ms latency on STM32H743IIT6. Comparative tests against single-modal and simple averaging baselines demonstrated accuracy and macro-F1 gains of over 2%. Zero-shot evaluations under bearing-wear and rotor-imbalance faults maintained ≥90.9% accuracy, while the D–S conflict and entropy measures enabled anomaly flagging with ROC-AUC 0.982. These results confirm the proposed framework's robustness, real-time deployability, and potential for extension to predictive maintenance and cross-motor generalization.

## Supporting information

**S1 Text. Paper program.**
(PDF)

## Author contributions

**Conceptualization:** Gong Chu.

**Data curation:** Gong Chu.

**Formal analysis:** Gong Chu, Zeng Peng.

**Funding acquisition:** Gong Chu, Zeng Peng.

**Investigation:** Gong Chu, Zeng Peng.

**Methodology:** Gong Chu, Zeng Peng.

**Project administration:** Gong Chu, Zeng Peng.

**Resources:** Gong Chu, Zeng Peng.

**Software:** Gong Chu, Zeng Peng.

**Supervision:** Zeng Peng.

**Validation:** Gong Chu.

**Visualization:** Gong Chu.

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
