## [Decision Letter · Decision Letter 0]

14 Aug 2025

PONE-D-25-33127Real-Time Motor Operating State Recognition via Multi-Sensor Fusion: A Wavelet–Neural–Evidence Framework for Industrial Condition MonitoringPLOS ONE

Dear Dr. Peng,

Thank you for submitting your manuscript to PLOS ONE. After careful consideration, we feel that it has merit but does not fully meet PLOS ONE’s publication criteria as it currently stands. Therefore, we invite you to submit a revised version of the manuscript that addresses the points raised during the review process.

We look forward to receiving your revised manuscript.

Kind regards,

Kannadhasan Suriyan

Academic Editor

PLOS ONE

Journal Requirements:

3. We note that your Data Availability Statement is currently as follows: All relevant data are within the manuscript and in Supporting Information files.

Additional Editor Comments:

AUTHOR SHOULD SUBMIT THE REVISED PAPER AS PER REVIEWER COMMENTS

Reviewers' comments:

Reviewer's Responses to Questions

**Comments to the Author**

1. Is the manuscript technically sound, and do the data support the conclusions?

Reviewer #1: Yes

Reviewer #2: Yes

2. Has the statistical analysis been performed appropriately and rigorously? 

Reviewer #1: Yes

Reviewer #2: Yes

3. Have the authors made all data underlying the findings in their manuscript fully available?

Reviewer #1: No

Reviewer #2: Yes

4. Is the manuscript presented in an intelligible fashion and written in standard English?

Reviewer #1: No

Reviewer #2: Yes

5. Review Comments to the Author

Reviewer #1: This paper presents a multi-sensor fusion framework for intelligent motor condition monitoring, which integrates wavelet-based feature extraction, shallow neural network classification, and evidence-theoretic decision fusion. The work is relatively thorough, but before the publication of the paper, further revisions are needed based on the feedback.

1.The innovation of this paper is not prominent. Wavelet packet transformation, BP neural network, and DS fusion are all mature methods. Although their combination has practical value, it lacks theoretical innovation and algorithm improvement.

2.In the first paragraph of the introduction, it is stated that motors are prone to faults such as bearing wear, rotor imbalance, insulation breakdown, and thermal stress. However, in the experimental design, only three states - "shutting down, no-load, and load" - were tested, without including the fault states. Please provide a reasonable explanation for this.

3.Please explain the reasons for the selection of wavelet packet decomposition parameters (as stated in "Signal Processing and Feature Extraction Method": the db4 wavelet was chosen for wavelet packet decomposition and a 3-level decomposition was performed). Also, conduct a sensitivity analysis of different parameters on the results.

4.In "State Recognition Model Construction", it is mentioned that "the model parameters are optimized using the Adam optimizer, with an initial learning rate of 10^-3, and following a decay schedule. An early stopping strategy based on validation loss is adopted to prevent overfitting and improve generalization ability". It is recommended to supplement the specific strategy for learning rate decay and the specific criteria for early stopping.

5.The title clearly includes "Real-Time Motor Operating State Recognition", while the abstract emphasizes "real-time monitoring" and "low-latency data acquisition". However, the full text does not demonstrate real-time performance. Please add this information.

6.There is a lack of detailed explanation regarding the selection of hyperparameters for the three-layer feedforward backpropagation neural network. Please provide the necessary information.

Reviewer #2: 1. Faulty conditions are not considered in the experiments.

2. Data samples are not sufficient. Also single motor data only is collected. Consider various motors for data collection.

3. Needs to perform various statistical analysis

4. Needs to do comparative analysi

6. PLOS authors have the option to publish the peer review history of their article (what does this mean?). If published, this will include your full peer review and any attached files.

Reviewer #1: No

Reviewer #2: No

---

## [Author Response · Author response to Decision Letter 1]

19 Aug 2025

We would like to thank the editors and reviewers for their constructive comments, which helped us to make significant improvements in the manuscript. In this regard, we provide point-by-point answers and amendments to the questions raised. All changes in the revised draft are marked in red and can be found in the attachment.

---

## [Decision Letter · Decision Letter 1]

19 Sep 2025

PONE-D-25-33127R1Real-Time Motor Operating State Recognition via Multi-Sensor Fusion: A Wavelet–Neural–Evidence Framework for Industrial Condition MonitoringPLOS ONE

Dear Dr. Peng,

Thank you for submitting your manuscript to PLOS ONE. After careful consideration, we feel that it has merit but does not fully meet PLOS ONE’s publication criteria as it currently stands. Therefore, we invite you to submit a revised version of the manuscript that addresses the points raised during the review process.

**ACADEMIC EDITOR: Minor revision**==============================

We look forward to receiving your revised manuscript.

Kind regards,

Agbotiname Lucky Imoize

Academic Editor

PLOS ONE

Journal Requirements:

Additional Editor Comments:

Reviewer #3: Revise the paper accordingly, but no not include the self-published papers suggested by the reviewer, especially the ones not related to the current study.

Reviewers' comments:

Reviewer's Responses to Questions

**Comments to the Author**

1. If the authors have adequately addressed your comments raised in a previous round of review and you feel that this manuscript is now acceptable for publication, you may indicate that here to bypass the “Comments to the Author” section, enter your conflict of interest statement in the “Confidential to Editor” section, and submit your "Accept" recommendation.

Reviewer #1: All comments have been addressed

Reviewer #3: All comments have been addressed

2. Is the manuscript technically sound, and do the data support the conclusions?

Reviewer #1: Yes

Reviewer #3: Yes

3. Has the statistical analysis been performed appropriately and rigorously? 

Reviewer #1: Yes

Reviewer #3: Yes

4. Have the authors made all data underlying the findings in their manuscript fully available?

Reviewer #1: No

Reviewer #3: Yes

5. Is the manuscript presented in an intelligible fashion and written in standard English?

Reviewer #1: Yes

Reviewer #3: Yes

6. Review Comments to the Author

Reviewer #1: (No Response)

Reviewer #3: This revised manuscript proposes a real-time condition-classification pipeline that extracts multiscale features with wavelet packet decomposition, feeds per-channel classifiers, and fuses decisions at the output. The authors also report embedded latency to support deployability. The revision is substantially improved and addresses most prior concerns. Before acceptance, I suggest the following focused clarifications and small additions.

- Clarify and guard against data leakage

You state that continuous recordings were windowed into 1024-sample segments with 50% overlap and then split 60/20/20. Please make explicit that the split is done at the recording/session (or subject) level—not at the window level—so that no windows from the same 10-minute session appear in both training and test sets.

- Provide a code/firmware repository link

You note dataset availability, but there is no link to the processing/training code and the embedded inference code. Public code greatly improves reproducibility and speeds up adoption.

Share a minimal GitHub repository (preprocessing, feature extraction, training, inference; plus a simple script to reproduce one main table/figure).

- Discuss complex and overcomplete wavelets as a natural extension

Your results motivate a short Discussion paragraph on why complex/overcomplete wavelet families (e.g., dual-tree complex, rational-dilation, or tunable Q-factor designs) may further improve shift-robustness, directional selectivity, and transient capture, potentially benefiting early-fault signatures.

Cite the following works:

https://doi.org/10.1109/ACCESS.2024.3524763

https://doi.org/10.1109/IEMBS.2011.6091193

https://doi.org/10.1109/EMBC.2014.6943877

https://doi.org/10.1007/s11517-014-1224-0

https://doi.org/10.1016/j.bspc.2010.09.007

https://doi.org/10.1016/j.compbiomed.2018.11.004

https://doi.org/10.1007/s11517-013-1114-x

7. PLOS authors have the option to publish the peer review history of their article (what does this mean?). If published, this will include your full peer review and any attached files.

Reviewer #1: No

Reviewer #3: No

---

## [Author Response · Author response to Decision Letter 2]

28 Sep 2025

We sincerely thank the reviewers and editors for their constructive feedback.

All comments have been carefully addressed in the revised manuscript.

Revisions are clearly marked in red in the text, and detailed point-by-point responses have been provided in the accompanying “Response to Reviewers” document.

We believe these changes have improved the clarity, rigor, and completeness of the paper, and we respectfully resubmit it for your further consideration.

---

## [Editor Report · Decision Letter 2]

9 Oct 2025

Real-Time Motor Operating State Recognition via Multi-Sensor Fusion: A Wavelet–Neural–Evidence Framework for Industrial Condition Monitoring

PONE-D-25-33127R2

Dear Dr. Peng,

We’re pleased to inform you that your manuscript has been judged scientifically suitable for publication and will be formally accepted for publication once it meets all outstanding technical requirements.

Kind regards,

Agbotiname Lucky Imoize

Academic Editor

PLOS ONE

Additional Editor Comments (optional):

Accept in current form.
---

## [Editor Report · Acceptance letter]

PONE-D-25-33127R2

PLOS ONE

Dear Dr. Peng,

I'm pleased to inform you that your manuscript has been deemed suitable for publication in PLOS ONE. Congratulations! Your manuscript is now being handed over to our production team.

Kind regards,

on behalf of

Mr. Agbotiname Lucky Imoize

Academic Editor

PLOS ONE